# A Multidisciplinary Approach as a Goal for the Management of Complications in Systemic Scleroderma: A Literature Review and Case Scenario

**DOI:** 10.3390/diagnostics13213332

**Published:** 2023-10-28

**Authors:** Dariana-Elena Pătrîntașu, Hédi Katalin Sárközi, Eugeniu Lupușor, Irina Elena Vlangăr, Gheorghe-Marian Rotariu, Ionuț-Alexandru Rența, Anda-Nicoleta Nan, Corina Eugenia Budin

**Affiliations:** 1Pneumology Department, Mures Clinical County Hospital, 540142 Targu Mures, Romania; dpatrintasu@yahoo.com (D.-E.P.); baloghedikatalin@yahoo.com (H.K.S.); lupusor.eugeniu@gmail.com (E.L.); ionutrenta@gmail.com (I.-A.R.); cora_bud@yahoo.com (C.E.B.); 2Pneumology Department, George Emil Palade University of Medicine, Pharmacy, Science and Technology of Târgu Mures, 540139 Targu Mures, Romania; 3Cardiology Department, Elias University Emergency Hospital, 011461 Bucharest, Romania; irina.bercea@yahoo.com; 4Faculty of Medicine, George Emil Palade University of Medicine, Pharmacy, Science and Technology of Târgu Mures, 540139 Targu Mures, Romania; rotariumarian9698@gmail.com; 5Pathophysiology Department, George Emil Palade University of Medicine, Pharmacy, Science and Technology of Târgu Mures, 540139 Targu Mures, Romania

**Keywords:** systemic sclerosis, interstitial lung disease, lung diseases, Raynaud’s syndrome

## Abstract

Systemic sclerosis (also known as scleroderma) is a chronic fibrosing autoimmune disease with both skin and multisystem organ involvement. Scleroderma has the highest mortality among all rheumatic diseases. The pathophysiology mechanism of systemic sclerosis is a progressive self-amplifying process, which involves widespread microvascular damage, followed by a dysregulation of innate and adaptive immunity and inflammation and diffuse fibrosis of the skin and visceral organs. Fibrosis of internal organs is a hint for systemic sclerosis, moreover associated with interstitial lung disease (SSc-ILD) is a complex process. In order to correlate scientific data from the literature with clinical experience, we present the case of a 56-year-old woman who was diagnosed with systemic sclerosis 16 years ago. The association of numerous comorbidities characterized by a considerable level of seriousness characterizes this case: the highly extensive systemic damage, the cardiovascular impact of the illness, and the existence of severe pulmonary arterial hypertension. The systemic and clinical manifestations, respiratory functional tests, radiological features, and specific therapy are discussed.

## 1. Introduction

Systemic sclerosis (also known as scleroderma) is a rare chronic fibrosing autoimmune disease with both skin and multisystem organ involvement [1,2]. Scleroderma has the highest mortality among all rheumatic diseases [3]. Its pathophysiology is complex; an altered balance of the acquired and innate immune system leads to the release of several cytokines and chemokines, as well as autoantibodies, which induce the activation of fibroblasts with the formation of myofibroblasts and the formation of a rigid connective tissue [1].

Fibrosis of the internal organs is a hint of systemic sclerosis, and when also associated with interstitial lung disease (SSc-ILD), it is a complex process involving inflammation, alveolar epithelial damage, and activation of resident fibroblasts, resulting in the thickening of the lung interstitium [2,4,5].

In this paper, we have conducted a case-based literature review of pulmonary fibrosis related to systemic scleroderma, because the increased risk of developing pulmonary fibrosis and pulmonary artery hypertension is among the leading causes of SSc-related deaths.

### 1.1. Epidemiology and Pathophysiology

Systemic sclerosis is a rare rheumatological condition, with a prevalence of 7.2–33.9, 13.5–44.3, and 8.2 per 100,000 individuals in Europe, North America, and The United Kingdom, respectively, and with an annual incidence of 0.6–5.6 per 100,000 individuals. Like many other rheumatological conditions, women are more commonly affected than men, with a reported ratio of between 3:1 and 8:1 [2,6].

Genetic factors likely contribute towards disease susceptibility and could explain some of the clinical heterogeneity of the disease. Also, environmental and occupational exposures, specifically silica, solvents, pesticides, and epoxy resins, have been implicated as potential causative factors [6,7,8].

The pathophysiology mechanism of systemic sclerosis is a progressive self-amplifying process, which involves widespread microvascular damage, which is believed to play a central role, followed by a dysregulation of innate and adaptive immunity and inflammation and diffuse fibrosis of the skin and visceral organs [6,9,10,11]. Most likely, vascular injury (possibly initiated by viruses, autoantibodies, chemicals, or oxidative products) and dysfunction of the endothelium cause local tissue ischemia, which promote tissue fibrosis [11].

Raynaud’s phenomenon is the clinical consequence of repeated vascular damage and vasospasm of the small arteries and arterioles of the fingers and toes, and it can be triggered by cold or even emotional stress. This is accompanied by an altered expression of adhesion proteins and cytokines [12].

The presence of serum autoantibodies that target multiple intracellular antigens distinguishes SSc. These autoantibodies, which are seen in more than 95% of patients, can be used to screen for SSc. ANA has been shown to occur in 75–95% of SSc patients, with an immunofluorescence sensitivity of 85% and a specificity of 54% [13]. 

Furthermore, anti-topoisomerase I (ATA) antibodies were previously known as anti-Scl-70 antibodies. ATA was found in 15–42% of SSc patients, with a sensitivity of 90–100%. The sensitivity of ATA is 34%. ATA is related to diffuse cutaneous SSc (dcSSc) and has a dismal prognosis. In SSc patients with ATA, the risk of severe lung fibrosis and cardiac involvement is enhanced. Additionally, ATA has been linked to the development of digital ulcers and joint involvement [13,14].

A fibroblast-to-myofibroblast transition is believed to be a key event and is driven by a number of profibrotic factors, in particular the transforming growth factor-beta [6,8].

The dysregulation of the innate and adaptive immune system response plays an important role and includes the increased presence and altered functions of inflammatory cells and products in target tissues, such as the skin and lungs, together with a polymorphism in IFN-regulatory factors that confers an increased risk of SSc [1,9,15,16].

The inflammatory profibrogenic cytokines and growth factors lead to the activation of fibroblasts [1,17]. The origin of these fibroblasts has been argued to derive from the circulation and also from the subcutaneous layer, from transdifferentiation, or resident cells in the tissue [1]. Activated fibroblasts produce ET-1, a potent vasoconstrictor that is able to increase fibronectin synthesis in normal and SSc human skin [9,18]. These have the characteristics of myofibroblasts, which have long been regarded as the key culprit in SSc fibrosis [17,19]. 

Another interesting aspect of SSc is the loss of subcutaneous adipose tissue [1,20]. Subcutaneous adipose mesenchymal stem cells and mature adipocytes are both involved in the transdifferentiation into fibroblast-like cells. The adipocytes from fibrotic locations in SSc are phenotypically different from normal adipocytes [21,22].

The progression of the pathophysiological pathways implicated in SSc is mirrored by the patient’s clinical features [23,24,25].

### 1.2. Diagnosis and Classification

Because there is no single diagnostic test, the diagnosis of systemic sclerosis is usually based on clinical features, but is supported through results from targeted investigations, following from which several sets of classification criteria have been developed such as the 2013 American College of Rheumatology/European League Against Rheumatism (ACR/EULAR) Criteria [6,26,27].

The most widely used technique divides systemic sclerosis into subsets: limited (LcSSC) and diffuse cutaneous systemic sclerosis (DcSSc), based upon skin involvement [28,29,30]. The term ‘CREST’ (calcinosis, Raynaud’s phenomenon, esophageal dysmotility, sclerodactyly, and telangiectasis) is a useful hint for some of the dominant features of systemic sclerosis, because patients with a diffuse disease can also develop all of these manifestations [6].

Moreover, very early systemic sclerosis should be suspected in the presence of RP (Raynaud’s phenomenon), PF (puffy fingers), and ANA positivity (positive antinuclear antibody), which were identified as the three “red flags” that should raise suspicion [31,32].

### 1.3. Clinical Manifestations

#### 1.3.1. Skin

Skin fibrosis is one of the dominant clinical features of SSc. The extent of skin fibrosis in SSc is most commonly assessed using the modified Rodnan skin score (mRSS), which measures skin thickness on a scale of 0 to 3 at 17 anatomical sites (score range 0–51) [33,34].

#### 1.3.2. Digital Vascular Disease

The fingers are commonly affected, but other sites can be involved too, including the toes and other vascular areas (e.g., lips and ears) [6]. Digital ulcers are a combination of progressive microangiopathy and digital artery disease, which are commonly observed in systemic sclerosis [35,36]. Often, they occur on the fingertips and over the dorsal (extensor) aspects of the hands, overlying the small joints [35].

#### 1.3.3. Cardiovascular System

In systemic sclerosis, cardiovascular involvement is common and can be life-threatening, because primary cardiac manifestation is often subclinical [37]. Arrhythmias are one of the most severe and potentially fatal complications of SSc, but an increased risk of atherosclerotic disease has also been reported [31,38].

#### 1.3.4. Respiratory Tract

Today, respiratory involvement (pulmonary fibrosis and pulmonary artery hypertension) is among the leading causes of SSc-related deaths [31,39]. Systemic-sclerosis-associated interstitial lung disease is the end result of the interplay between fibrosis, autoimmunity, inflammation, and vascular injury [40]. Clinical symptoms occur late and are nonspecific, but when reported, dyspnea, nonproductive cough, and overwhelming fatigue are the most common symptoms [40,41]. A physical examination reveals Velcro-like crackles on auscultation in addition to the cutaneous findings, and a pulmonary function evaluation often reveals restriction [40,42,43]. 

HRCT is the gold standard for the early diagnosis of SScILD [4,42]. The most common imaging pattern on HRCT is nonspecific interstitial pneumonia, which is seen in more than 70–80% of patients with SSc-ILD [40,42,44]. It is characterized by peripheral ground-glass opacities with an apical to basal gradient, frequently accompanied by subpleural sparing. Parenchymal changes are defined by the presence of reticulation, traction bronchiectasis, and bronchiolectasis in a similar distribution [42,45]. Also, functional pulmonary testing (spirometry and Dlco), is mandatory to identify patients with developing progressive interstitial lung disease.

#### 1.3.5. Gastrointestinal and Renal System

An increased deposition of collagen and other components of the extracellular matrix leads to fibrotic changes in the upper and lower GI tract, resulting in dysmotility, malabsorption, malnutrition, and dilation of the intestine [46,47]. The commonly reported symptoms of SSc include meteorism (87%), fecal incontinence (23%), and features related to reduced esophageal motility or gastroparesis like dysphagia, heartburn, and gastrointestinal reflux symptoms [6,46]. In addition to the pulmonary clinical manifestations, coughing and a sore voice can occur [47,48].

Moving forward, among all possible systemic sclerosis internal organ complications, kidney involvement is frequently underestimated, because it is usually attributed to other health problems [49,50]. The primary event of kidney damage is an injury to the endothelial cells, causing an intimal thickening and proliferation of the intralobular and arcuate arteries [50]. Typical clinical features appear with the onset of accelerated hypertension: severe headache, blurred vision, and other encephalopathic symptoms. Otherwise, most patients with scleroderma renal crisis (SRC) complain of nonspecific symptoms, including increased fatigue, dyspnea, or just dizziness [51]. 

#### 1.3.6. Musculoskeletal System

The musculoskeletal system is commonly affected in patients with systemic sclerosis [6]. For example, joint involvement can range from nonspecific arthralgia and myalgia to rheumatoid arthritis (RA) [52]. Hand (finger) flexion contractures are an important cause of disability [53]. Bilateral carpal tunnel syndrome (CTS) or median neuropathy at the wrist (MNW) can be seen in patients with early disease and is sometimes the first non-Raynaud’s presenting feature of systemic sclerosis [52,54]. 

## 2. Case Scenario

The presented case is from a patient admitted to the Pneumology Department of the Mures Clinical County Hospital. Informed consent was obtained from the patient. This study was conducted in accordance with the Declaration of Helsinki. 

We present the case of a 56-year-old woman with prolonged professional exposure (worked in vulcanization), diagnosed with systemic sclerosis 16 years ago (anti-Scl-70 antibodies = 7.2, antinuclear antibodies = 21.3 UI/mL). Onset of symptoms occurred in 2003, and onset of secondary Raynaud’s syndrome, digital ulcers, recurrent pneumonia, and pericarditis occurred in 2007. Additionally, the patient experienced respiratory failure and an interstitial lung disease progressive pattern, and was initially on treatment with immunosuppressive agents (Cyclophosphamide 100 mg/day) (stopped in December on her own initiative) and later on treatment with Methotrexate 20 mg/week (stopped in June 2012 for administrative reasons). Since December 2015, treatment with endothelin receptor antagonists (Bosentan, Bosentan-Manufacturer Actavis, 2 × 125 mg/day) has been initiated, to which a PDE-5 inhibitor (Sildenafil, Sildenafil-Manufacturer Terapia Iași, Romania 3 × 20 mg/day) has been associated since 2019, with the patient being included in the National Treatment Program for patients with Pulmonary Arterial Hypertension. On 7 May 2022, antifibrotic treatment with tyrosine-kinase inhibitors was initiated (Nintedanib, Nintedanib-Manufacturer Boehringer Ingelheim, Romania 150 mg 2 × 1/day, later 100 mg 2 × 1/day from 12 December 2022). The patient is also known to have viral hepatitis B (under antiviral treatment with Entecavir, Entecavir-Manufacturer Labormed, Romania), L4–L5 spondylolisthesis, cervical-dorso-lumbar spondylodiscarthrosis, hepatic steatosis, reflux esophagitis, pangastric erosive disease, and hemorrhoidal disease (upper digestive endoscopy and colonoscopy 2018). Moreover, she was recently diagnosed with osteoporosis (T-score −3.1) and low serum levels of vitamin D and was initiated on treatment with bisphosphonate medication (Ibandronic Acid 3 mg/day) and high doses of D3 vitamin. 

Objective examination and examination of the locomotor system:

A normosthenic patient, BMI 23.78 kg/m^2^, face with the appearance of a “Byzantine icon”, widened palpebral fissures with deletion of nasolabial folds, decreasing mouth opening and multiplication of peribuccal folds. Indurated, hyper-pigmented skin, sclerodactyly, fingers fixed in flexion, bilateral upper limb digital ulcerations, II, III, V toes, multiple stellate scars at the level of the phalanges of the upper limbs, cold, pale feet with an active ulcer in the second toe of the left foot. Active mobilization is accompanied by diffuse joint pain and cramping when mobilizing the knees, accentuated dorsal kyphosis, ante projected shoulders, limitation of lateral flexion, cervical and lumbar spine pain, flattening of the lumbar lordosis, muscle contracture of the upper border trapezium and bilateral dorso-lumbar paravertebral, percussive sensitivity of the spinous apophyses dorso-lumbar spine, painful limited ante flexion, crural plexus elongation positive bilaterally, Schober 10/13 cm, Lasegue positive bilaterally. Kyphotic thorax, vesicular murmur present bilaterally, bilateral basal Velcro rales, oxygen saturation 87% in ambient air, 97% with oxygen mask 6–8 L/min. Apexian shock in the left V intercostal space on the medio-clavicular line, rhythmic heart sounds, sound II accentuated and doubled at the pulmonary area, ventricular allure 68/min, blood pressure 110/80 mmHg, palpable peripheral pulse bilaterally at the pedis artery. Microstomia, abdomen located in the xipho-pubic plane, sensitive to deep palpation in the right hypochondrium, the spleen is not palpable, the Giordano maneuver is negative bilaterally. Temporospatially oriented, symmetrical triggerable osteo tendinous reflexes, without signs of meningeal irritation, overall low muscle strength, low tactile and superficial sensitivity at the level of the affected skin.

### 2.1. Investigations

#### 2.1.1. Laboratory Investigations (Table 1)

#### 2.1.2. Electrocardiography (Figure 1)

Sinus rhythm, right axis deviation, right bundle branch block (RBBB), negative T-wave in DIII, V1–V4.

**Figure 1 diagnostics-13-03332-f001:**
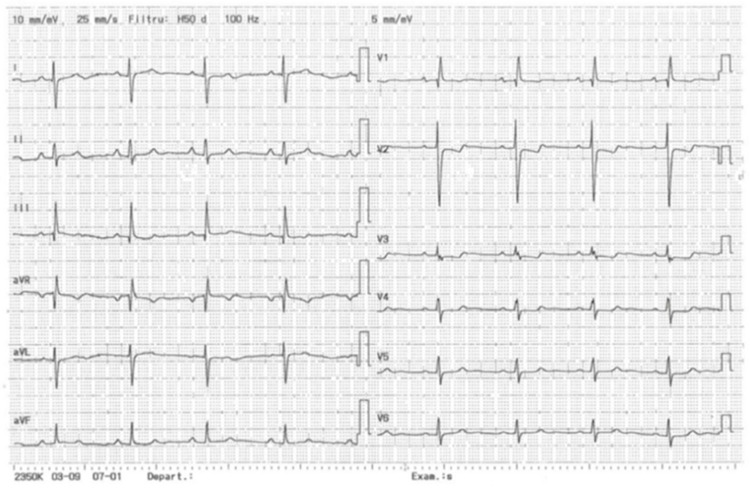
Electrocardiogram of the patient.

#### 2.1.3. Minutes Walking Test

At the start, ventricular rate = 59/min and oxygen saturation is 85% in ambient air, dyspnea (Borg scale) = 2. 

At stop, ventricular rate is 71/min, saturation is 82%, dyspnea (Borg scale) = 7. Total distance = 50 m (9% of the predicted distance = 551 m). The test was stopped after one minute and 13 s, due to marked dyspnea and the feeling of vertigo manifested by the patient.

#### 2.1.4. Echocardiography (Table 2)

Mitral valve: mobile, moderate atherosclerotic changes, hemodynamically insignificant. Aortic valve: tricuspid, mobile, moderate ATS changes, hemodynamically insignificant. Tricuspid valve: normally inserted, flexible, mobile. Pulmonary valve: supple, mobile. Interatrial septum/interventricular septum: intact. Left ventricle: efficient, without segmental and global kinetic disorders, global EF 55%, diastolic dysfunction grade I. Right cavities: RV 30/34/56 mm, hypokinetic TAPSE = 17 mm, MAPSE = 13 mm. 

Conclusions: Efficient, nondilated LV, global EF 55% with diastolic dysfunction grade 1, minor mitral regurgitation, moderate pulmonary regurgitation. Hypokinetic RV. Severe systolic pulmonary hypertension, echocardiographic criteria showing very high probability of pulmonary hypertension. GLPS LAX: 11.5; GLPS A4C: 10.3; GLPS A2C: 13.1; AVERAGE GLPS: 11.7; Biplane FE: 55%; EDV: 72 mL; ESV 41 mL; SV 31 m; LVCO: 2.3 L/min; GLPS VD: 7.8%.

**Table 2 diagnostics-13-03332-t002:** Echocardiography of the patient.

Dimensions	Values in mm	Normal Values
RV	32	22–44
LV	36	35–60/21–40
IVS	11	6–11
Post. wall	10	6–11
LA	32	23–45
LA dimension in cm^2^	15	
LA volume	-	
RA	-	
RA dimension in cm^2^	16	
Ao ring	19	14–26
Ascendent Ao	23	21–34
PAring	21	10–22
PAtrunk	22	9–29
EF	55%	60%

RV: right ventricle; LV: left ventricle; IVS: interventricular septum; LA: left atrium; RA: right atrium; Ao: aortic; PA: pulmonary artery; EF: ejection fraction.

#### 2.1.5. Bone Densitometry (Table 3)

The BMD measured at the right femur neck is 0.601 g/cm^2^, with a T score of −3.1, as can be seen in Table 2. 

**Table 3 diagnostics-13-03332-t003:** Densitometry evaluation.

Densitometry Trend: Total Mean				
Measured Date	Age (Years)	BMD (g/cm^2^)	Change vs. Previous (g/cm^2^)	Change vs. Previous (%)
28 October 2022	55.4	0.693	−0.138	−16.6
19 June 2018	51.0	0.831	−0.031	−3.6
12 July 2016	49.1	0.862	-	-
Osteoporosis: YA t-score: −3.1				

#### 2.1.6. High-Resolution Computed Tomography (12 October 2022) (Figure 2)

Thyroid gland of normal appearance. Advanced fibrotic changes in both lung fields with septal thickening, architectural disorganization, and traction bronchiectasis, with the changes being more important at the basal level of the bilateral lower lobes. Small diffuse calcified granulomas bilaterally, without areas of pulmonary condensation. Absence of suspicious pulmonary nodules. Trachea and bronchi with free lumens. Absence of mediastinal masses. Absence of pleural fluid accumulations. Mediastinal adenopathies up to 17 mm perivascular, 16 mm pretracheal at the right, 18 mm left hilar, 15 mm right hilar, and multiple subcentimeter, some with punctate calcifications. Esophagus minimally dilated with liquid content. Cardiomegaly, pericardial blade up to 18 mm in the right ventricle. Accentuation of dorsal kyphosis. Early degenerative changes in the dorsal spine, without suspicious lesions on the scanned bone segment. Conclusions: Pulmonary fibrosis changes with medium-advanced damage. Pericardial minimum. Esophageal stasis, more likely in the context of achalasia. Bilateral mediastinal and hilar adenopathies, some with calcifications.

**Figure 2 diagnostics-13-03332-f002:**
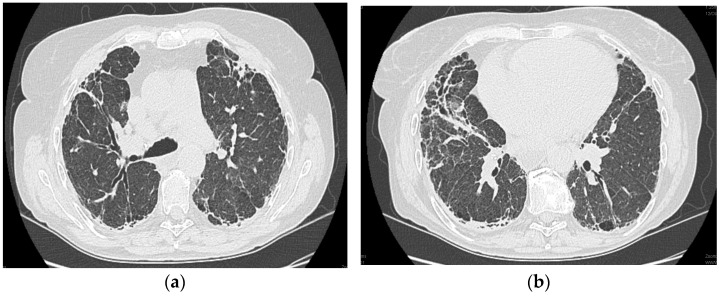
(**a**,**b**): Computed tomography of the thorax.

#### 2.1.7. Spirometry

Mixed ventilatory dysfunction predominantly restrictive, decreased vital capacity (VC) by 45%, decreased forced expiratory volume in one second (FEV1) by 45.4%. Normal Tiffneau index.

#### 2.1.8. Bodypletismography (Table 4)

Airway resistance (RAW) and total airway resistance (Rtot) normal values, resistance–volume ratio (R-V) 83% (normal value), low total lung capacity of 67% (TLC) and functional residual capacity (FRC) of 80%.

**Table 4 diagnostics-13-03332-t004:** Bodypletismography.

Bodyplethysmography/Flow Volume			
	Pred	Pre	% (Pre/Pred)
R tot	0.30	0.35	116
sG tot	1.04	1.10	106
R eff	0.30	0.28	93
FRCpl	2.75	2.19	80
RV	1.87	1.55	83
TLC	5.10	3.4,	67
VC IN	3.09	1.44	47
FVC	2.99	1.87	63
FEV 1	2.54	1.46	57
FEV1%M	78.65	77.87	99
FEV1%F	78.65	77.87	99
PEF	6.32	3.54	58
FEV6		4	
MEF 75	5.54	2.57	46
MEF 50	3.63	1.74	45
MEF 25	1.47	0.55	37

#### 2.1.9. Lung Diffusion Capacity (Table 5)

Very low values of diffusing capacity of the lung for carbon monoxide (Dlco) (25%) and carbon monoxide transfer coefficient (Kco) (41%).

**Table 5 diagnostics-13-03332-t005:** DLCO.

Diffusion SB			
	Pred	Best	%(Best/Pred)
DLCO_SB mmol/(min·kPa)	8.06	2.04	25
KCO_SB mmol/(min·kPa·L)	1.58	0.65	41
VA_SB (L)	4.95	3.12	63
Hb g(Hb)/dL	13.50	13.40	99
DLCO mmol/(min·kPa)	8.06	2.04	25
KCOc_SB mmol/(min·kPa·L)	1.58	0.65	41
VIN (L)		0.00	
TLC_SB (L)	5.10	3.26	34
FRC_SB (L)	2.75	2.20	90
ERV_SB (L)	0.88	0.72	91
RV_SB (L)	1.87	1.48	90
RV%TLC_SB (%)	38	46	121

Functional respiratory studies reveal a restrictive-type pattern due to pulmonary fibrotic lesions. Pulmonary interstitial damage is caused by the rheumatological pathology, progressive systemic scleroderma coexisting with the etiological context of pulmonary arterial hypertension.

#### 2.1.10. High-Resolution Computed Tomography (March First 2023) (Figure 3)

Predominantly subpleural reticular lesions, with a four-cornered appearance, associated with minimal right anterobasal peribronchovascular extension. Traction bronchiectasis is associated with the reticular beaches above. Subpleural areas of honeycombing are more accentuated in the lower half of the lung. Discrete peripheral organizing masses, especially in the left posterior. Fibrous bands with small associated calcifications. The pleural contour is irregularly marked, with numerous spicules on the contour. The esophagus is markedly dilated along its entire length, with a caliber of up to 26 mm, regular walls, and liquid stasis in the lower half. Circumferential pericarditis in a small amount. Global cardiomegaly. The pulmonary artery cone has a caliber of 36 mm, in the context of known pulmonary hypertension. Numerous supracarinal bilateral mediastino-hilar adenopathies, some of them with small calcifications, with an inflammatory appearance. Thyroid with normal position and dimensions, inhomogeneous, micropolynodular structure. Conclusions: The CT appearance is an appearance of interstitial lung pneumopathy: progressive fibrosing phenotype, examination quasi-identical to the previous CT examination. Dilated pulmonary arteries with the appearance of PAH. Minimal pericarditis. Cardiomegaly. Polynodular goiter. Dorsal spondylarthrosis.

**Figure 3 diagnostics-13-03332-f003:**
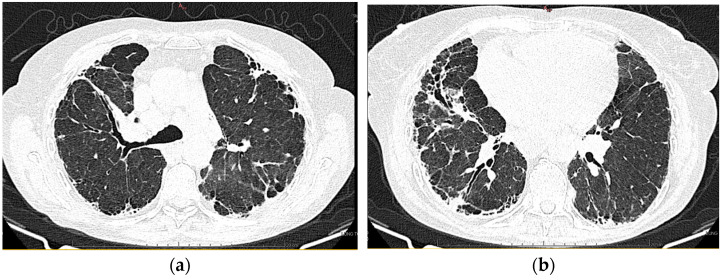
(**a**,**b**): Reevaluation of lung tomography.

Considering the underlying pathology, progressive systemic sclerosis and interstitial lung disease, the next step in the diagnostic process was progressive evaluation. The restrictive pattern expressed on the body plethysmography associated with a significant decrease in DLCO, the clinical deterioration of the patient, and the progression of imaging lesions on HRCT led to the classification of the patient as having an SSC-ILD progressive phenotype.

### 2.2. Management of SSc

The treatment of SSc can be tough due to its rarity and heterogeneous disease manifestations [55]. Best practice often involves shared medical care and therapy, which should be implemented to aim directly at active organ-specific complications of the disease.

#### 2.2.1. Cutaneous and Vascular Involvement

In addition to skin thickening, a cutaneous disease involves the presence of calcinosis and pruritus, which are common results as a consequence of small fiber neuropathy [56]. Immunosuppressive therapies include methotrexate and mycophenolate mofetil, with the modified Rodnan skin score (mRSS) routinely used to quantify the extent of cutaneous sclerosis [55,57]. For the peripheral vascular system (Raynaud’s phenomenon, digital ulcers, and critical ischemia), the following therapies help reduce the frequency and severity of vascular manifestations: calcium channel blockers, phosphodiesterase type 5 inhibitors, angiotensin II receptor blockers, endothelin receptor antagonists, prostacyclin analogue, wound care for digital ulcers, and antibiotic therapy for infected ulcers [6,55].

#### 2.2.2. Heart involvement

Heart involvement is a strong prognostic factor in systemic sclerosis and may present more frequently with diastolic (rather than systolic) dysfunction (heart failure with preserved ejection fraction) [56,58]. Current pharmacological therapies for heart failure include the usual drug therapies such as calcium channel blockers for prevention and treatment of left ventricular dysfunction, ACE inhibitors, diuretics, or calcium channel blockers for improvement in myocardial perfusion and anti-arrhythmic agents [6,55]. Regarding the inflammatory cardiac profile, immunosuppressive therapy (for example corticosteroid or cyclophosphamide drugs) should be taken into consideration.

#### 2.2.3. Scleroderma Renal Crisis

The use of ACEI to treat SRC has been associated with a good outcome and is mandatory for the improvement in morbidity and mortality due to scleroderma renal crisis [59]. Additionally, education for those at high risk regarding a proper routine of monitoring blood pressure and close communication of new symptom development (headache, dyspnea, dizziness, syncope) is strongly recommended [56].

#### 2.2.4. Gastrointestinal Disease

The right management is based on the symptoms appearance and evolution, and it includes a proton pump inhibitor or H2 blockers for esophageal acid reflux disease, nutritional supplementation for those with a restricted diet or malabsorption, and esophageal dilatation for persisting dysphasia [55,56].

#### 2.2.5. Interstitial Lung Disease and Pulmonary Hypertension

Excepting methotrexate, immunosuppressive treatments for cutaneous fibrosis are frequently successful for treating SSc-ILD, highlighting the similar etiology of both symptoms [60]. The recommendations are for cyclophosphamide, mycophenolate, rituximab, or tocilizumab, with priority given to mycophenolate due to its documented effectiveness for interstitial lung disease and skin and its good side effect profile [61,62]. Also, the tyrosine kinase inhibitor Nintedanib is authorized for use in treating progressive pulmonary fibrosis [61].

The progression of the interstitial pathology in cases with scleroderma is a serious factor that impacts the prognosis of these patients, and it is similar to an amiodarone-induced lung injury [63]. Progressive pulmonary fibrosis (PPF) is defined as the presence of pulmonary fibrosis, to which two of the following criteria must be added: aggravation of respiratory symptoms; progression of the disease from a functional point of view (decrease in FVC > 5% predicted since the previous visit or in the last year or decrease in DLCO (corrected for Hb) > 10% since the previous assessment); or imaging evidence of disease progression evidenced on HRCT [64].

Cases in which, from a pneumological point of view, the presence of progression has been established, require special monitoring that requires the following investigations: repeating the functional tests and the walking test every 4–6 months, or sooner if the symptomatology requires it; repeating HRCT at 1 year, or less if there is another suspected diagnosis; performing Angio CT if there are signs of a pulmonary embolism.

All patients with SSc are at risk of developing pulmonary arterial hypertension. Phosphodiesterase 5 inhibitors (such as Sildenafil and Tadalafil), endothelin receptor antagonists (such as Bosentan or Ambrisentan), and prostacyclins are all used to attain functional New York Heart Association Class II or higher (light breathlessness) and minimal restriction when performing routine tasks [55,56].

#### 2.2.6. Musculoskeletal Involvement

Common sites for inflammatory arthritis found in systemic sclerosis include the hands, wrists, elbows, knees, and ankles [53]. Inflammatory arthritis symptoms may benefit from the use of low-dose corticosteroids (less than 10 mg/day) [56].

## 3. Discussion

Improving the management of this potentially fatal consequence of SSc requires early detection to stratify risks, monitor progression, and act when appropriate [61].

Considering that HRCT is the gold standard for detection of ILD, in our case, the patient was examined every 6–12 months, but only 50–66% of medical experts frequently conduct HRCT in newly diagnosed SSc patients; this demonstrates the wide diversity in global practice [43,65]. 

The peak age of onset is 55–69 years, and women are more commonly affected than men, [6] and while the full spectrum of SSc is seen among those with late-age onset SSc, as it can be seen in Manno et al.’s study [66], our patient, unfortunately, had an early onset of her clinical manifestations (at 36 years old), with cutaneous, pulmonary, cardiac, and gastrointestinal involvement.

Moinzadeh et al., in their cohort study from Germany (2020), imply that pulmonary hypertension and cardiac involvement occurred substantially more frequently within the late-onset sample in terms of organ manifestation, which is consistent with prior publications [67]. Also, Veronika K. Jaeger et al., in the largest direct comparison of different ethnicities from the EUSTAR 2004–2018 database strengthen the knowledge about the clinical and serological differences between Black, Asian and white people, in which Asian people had a higher prevalence of pulmonary hypertension and severe lung involvement [68].

Regarding our case, it is well known that since December 2015, the patient (Caucasian woman) has been included in the Romanian National Treatment Program for patients with Arterial Hypertension Pulmonary at the age of 48 years, with recurrent pneumonia and pericarditis since 2007.

It has been shown that late-age onset SSc was surprisingly protective against digital ischemia, with early age being previously described as a risk factor for digital ulcers in systemic scleroderma [66,67]. Our case fits this description, with digital ulcers involvement being present since the early onset in 2007. 

The EULAR Scleroderma Trial and Research cohort revealed 6.6% of deaths from SSc that resulted from gastrointestinal complications among elderly patients and patients with diffuse skin involvement [46,67]. The fibrosis of oral and perioral tissues, chronic inflammation, deformity of the oral cavity, and misalignment of osseous structures that result in microstomia and malocclusion of the teeth are the causes of oropharyngeal problems [46]. As a result, our patient suffered from the gastrointestinal symptomatology detailed in the scientific literature, including impaired mastication and deglutition, food leakage, regurgitation, and hoarseness of voice. Furthermore, approximately 50% to 90% of patients with scleroderma experience esophageal manifestations such as acid reflux, which further triggers erosive esophagitis [69,70]. Our patient underwent an upper digestive endoscopy in response to the symptoms mentioned above, and erosive esophagitis and chronic gastritis were ultimately diagnosed. Because of this gastroenterological involvement, a digestive intolerance to the initial dose of antifibrotic medication could be justified. The low dose of 200 mg per day divided into two doses could be tolerated without adverse effects.

Both the treatment of PAH and the treatment of esophagitis were managed in this case according to the latest European recommendations. The French practical guidelines brought updates to the ATS/ERS recommendations for the management of these comorbidities [71].

The connection between SSc and the risk of osteoporotic fracture did not reach statistical significance so far, according to Chen et al. in their meta-analysis study (2019). However, patients with gastrointestinal involvement have impaired vitamin D absorption, which leads to malnutrition and thickening of the skin or mucosa in SSc patients and reduces UV penetration and lowers pre-vitamin D3 production, leading to a lower overall bone density [72]. Outlining the points made above, our case demonstrated how malabsorption and skin thickening resulted in secondary osteoporosis, with a T score of −3.1; thus, this patient is considered osteoporotic according to the World Health Organization (WHO) criteria [73]. Moreover, vitamin D may interfere with each of the pathobiological processes triggered in SSc, including autoimmunity, peripheral vasculopathy, and fibrosis, due to its immunomodulatory, cardioprotective, and antifibrotic biological actions. Vitamin D levels appear to be considerably reduced compared with healthy controls, and vitamin D supplementation is mandatory [74].

Interstitial lung disease (ILD) and pulmonary arterial hypertension (PAH), the most common pulmonary symptoms of SSc, have been highlighted by clinical practice and the scientific community as the leading causes of death [75]. When compared with SSc patients without ILD, patients with SSc and ILD (SSc-ILD) had a mortality risk that was almost three times higher. Additionally, patients with Scl-70 (anti-topoisomerase I) antibodies, male sex, and African American race have a propensity for more severe SSc-ILD and a higher chance of worsening over time [75,76]. Despite the fact that our case does not fit the gender and ethnicity requirements described above, the presence of pulmonary fibrosis and pulmonary hypertension, as well as the presence of anti-Scl-70 antibodies that are positive, place the patient at a high risk of morbidity and mortality.

The recent European guidelines for the treatment of SSC-ILD support as a consensus the effectiveness of medication with Mycophenolate Mofetil, Cyclophosphamide, and Nintedanib. After identifying the progression, it is recommended to escalate the drug treatment, along with the nonpharmacological adjuvant. According to this algoritm based on a modified Delphi process, the patient in the presented case received pharmacological treatment with Nintedanib for ILD, as well as oxygen therapy at home. Immunomodulatory and antifibrotic therapies act mainly on the pathways related to autoimmune or inflammatory processes, respectively, and the pathways related to the production of fibrosis. Even if recent evidence suggests that immunomodulatory medication can also influence the appearance of fibrosis, self-sustaining pulmonary fibrosis requires an effective antifibrotic agent, and in the case of scleroderma, Nintedanib is the only antifibrotic licensed [77]. According to these recent recommendations, the patient in the presented case was initiated on antifibrotic therapy with Nintedanib.

Goh et al. demonstrated that patients with SSc-ILD had a greater risk of eventual death for FVC declines of 10% or for FVC declines of 5–9%, combined with a fall of 15% in Dlco [78], as indicated by our patient’s repeated respiratory functional tests. Nevertheless, while DLco has been found to be the strongest predictor of HRCT-measured ILD [76], Ryerson et al., additionally, identified that the 6 min walk test distance (6MWT) is an independent predictor of mortality [78]. Taking into consideration that the 6MWT is frequently used as a measure of exercise tolerance in patients with SSc-ILD [79,80], we tried to perform the test on our patient, but unfortunately, we have been forced to stop the examination after only one minute and 13 s, with no more than 9% of the predicted distance completed, because of the exacerbation of dyspnea and vertigo.

The particularity of this case consists of the association of multiple comorbidities characterized by a significant level of seriousness. The extremely important systemic damage, the cardiovascular impact of the disease with the presence of severe PAH, significantly affects the patient’s quality of life. The association with Hepatitis B limits the possibility of using immunosuppressive medications and, at the same time, excludes the last step of treatment escalation: lung transplantation.

## 4. Conclusions

In conclusion, because of its severe circulatory and pulmonary dysfunction, unexpected onset and course, and wide range of clinical manifestations, SSc is an overall challenging condition. Patient demographics, SSc-specific traits such as skin distribution and illness duration, serological markers, pulmonary function tests, and the degree of lung damage on HRCT are all important factors in risk factors for morbidity and mortality. Thus, to emphasize the impact of the interstitial lung involvement in systemic scleroderma, we have presented a narrative review and a case from a patient admitted to our Pneumology Department from Mures Clinical County Hospital, who was diagnosed with pulmonary fibrosis related to systemic scleroderma. By understanding the mechanism of developing progressive pulmonary fibrosis related to systemic scleroderma, a personalized treatment can pe established to improve the patient’s outcome and, through this, to increase the quality of life. 

## Figures and Tables

**Table 1 diagnostics-13-03332-t001:** Laboratory Analysis.

Parameter	Value
White blood cells	4630/µL
Hemoglobin	12.5 g/dL
Hematocrit	38.8%
Platelets	254,000/µL
Uric acid	303 µmol/L
Alanine transaminase	12 U/L
Aspartate aminotransferase	20 U/L
Total cholesterol	4.37 mmol/L
Creatinine	0.82 mg/dL
Alkaline phosphatase	210 U/L
Glucose	124 mg/dL
Lactate dehydrogenase	233 U/L
C-reactive protein	5.13 MG/L
Fibrinogen	345 mg/dL
Total proteins	7.2 g/dL
Anti-Scl-70 antibodies	7.2
Antinuclear antibodies	25.3 UI/mL

## Data Availability

Data is contained within the article.

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
