# Peer review of "A Multidisciplinary Approach as a Goal for the Management of Complications in Systemic Scleroderma: A Literature Review and Case Scenario"

_diagnostics, 2023, doi:10.3390/diagnostics13213332_

Round 1
Reviewer 1 Report
Comments and Suggestions for Authors
The manuscript is interesting and quite well written. I have several suggestions:
1- Systemic sclerosis (also known as scleroderma) is a chronic fibrosing autoimmune disease with both skin and multisystem organ involvement. Scleroderma has the highest mortality among all rheumatic diseases. The pathophysiology mechanism of systemic sclerosis is a progressive self-amplifying process, which implicates the widespread microvascular damage, followed by a dysregulation of innate and adaptive immunity and inflammation, and diffuse fibrosis of the skin and visceral organs. Fibrosis of internal organs is a hint for systemic sclerosis, moreover associated with interstitial lung disease (SSc-ILD) is a complex process. We report a case of a 56 years old female, diagnosed with Systemic Sclerosis 16 years ago. The systemic and clinical manifestations, respiratory functional tests, radiological aspects and specific therapy were discussed. Please underline the peculiar characteristics of your clinical case to attract the interest of possible readers and increase the citation index
2- Introduction. L44-52.The pathophysiology mechanism of systemic sclerosis is a progressive self-amplifying process, which implicates the widespread microvascular damage, which is believed to play a central role, followed by a dysregulation of innate and adaptive immunity and inflammation, and diffuse fibrosis of the skin and visceral organs [6,9,10,11]. Most likely vascular injury (possibly initiated by viruses, autoantibodies, chemicals, or oxidative products) and dysfunction of the endothelium causes local tissue ischaemia which promotes tissue fibrosis [11]. Fibroblast to myofibroblast transition is believed to be a key event, and is driven by a number of profibrotic factors, in particular transforming growth factor-beta [6,8]. I suggest to improve this part to complete the manuscript. Below you can find some works that could give useful ideas in expanding this part: I suggest to add these refences:
a-Advanced Autoantibody Testing in Systemic Sclerosis. Diagnostics 2023, 13, 851. https://doi.org/10.3390/diagnostics13050851
b- Progression of nailfold microvascular damage and antinuclear antibody pattern in systemic sclerosis. J Rheumatol. 2013 May;40(5):634-9. doi: 10.3899/jrheum.121089.
c- Innovations in the Assessment of Primary and Secondary Raynaud's Phenomenon. Front Pharmacol. 2019 Apr 16;10:360. doi: 10.3389/fphar.2019.00360.
3- Figure 2. a), b): Computed Tomography of the Thorax. Please, improve the quality of the figures.
4- Figure 5. a), b): Reevaluation of Lung Tomography. Please, improve the quality of the figures.
5- 3. Discussion and Conclusions: L346-351. Improving the management of this potentially fatal SSc consequence requires early detection to risk stratify, monitor progression, and act when appropriate [58]. Considering HRCT the gold standard for detection of ILD, in our case the patient was examined every 6-12 month, but only 50%-66% of medical experts frequently conduct HRCT in newly diagnosed SSc patients, this demonstrates the wide diversity in global practice [40,62]. Please, summarise here the novelty of the study.
6- L440-444. In conclusion, because of its severe circulatory and pulmonary dysfunction, unex- pected onset and course, and wide range of clinical manifestations, SSc is an overall challenging condition. Patient demographics, SSc-specific traits such skin distribution and illness duration, serological markers, pulmonary function tests, and the degree of ung damage on HRCT are all important factors in risk factors for morbidity and mortality. Please, summarise here the novelty of this paper and the possible clinical implications.
Comments on the Quality of English Language
Minor changes of English language are required
Author Response
Dear Reviewers, thank you very much for taking the valuable time to review our work and for providing your feedback, which I find very useful in significantly improving our article. The authors have carefully noted all coments and guidance and we tried our best to provide point-by-point answers to your observations along with the obligatory explanations. We have highlighted the changes within the manuscript via Track-Changes.
Response to Reviewer 1 Comments:
Point 1: Please underline the peculiar characteristics of your clinical case to attract the interest of possible readers and increase the citation index.
Response 1: From L22 to L27 we tried to underline the particularity of our case.
“In order to correlate scientific data from the literature with clinical experience, we present a case of a 56 year-old female., we report a case of a 56 year-old female, who was diagnosed with Systemic Sclerosis 16 years ago. The association of numerous comorbidities burdened by a considerable gravity characterizes this case: the highly extensive systemic damage, the cardiovascular impact of the illness and the existence of severe pulmonary arterial hypertension. The systemic and clinical manifestations, respiratory functional tests, radiological features and specific therapy were discussed.”
Point 2: Introduction. L44-52. I suggest to improve this part to complete the manuscript. Below you can find some works that could give useful ideas in expanding this part: I suggest to add these refences.
Response 2: Thank you for your indications, from L59 to L75 we included a more extensive description of the pathophysiological mechanisms consistently with the three references suggested.
“Raynaud’s phenomenon is the clinical consequence of repeated vascular damage and vasospasm of the small arteries and arterioles of the fingers and toes and it can be triggered by cold or even emotional stress. This is accompanied by an altered expression of adhesion proteins and cytokines.[12]. The presence of serum autoantibodies that target multiple intracellular antigens distinguishes SSc. These autoantibodies, which are seen in more than 95% of patients, can be used to screen for SSc. ANA has been shown to occur in 75-95% of SSc patients, with an immunofluorescence sensitivity of 85% and specificity of 54% [13]. Furthermore, anti-topoisomerase I (ATA) antibodies were previously known as anti-Scl-70 antibodies. ATA was found in 15-42% of SSc patients, with a sensitivity of 90-100%. The sensitivity of ATA is 34%. ATA is related with diffuse cutaneous SSc (dcSSc) and has a dismal prognosis. In SSc patients with ATA, the risk of severe lung fibrosis and cardiac involvement is enhanced. Additionally, ATA has been linked to the development of digital ulcers and joint involvement [13,14].”
Point 3 and 4: Figure 2. a), b): Computed Tomography of the Thorax and Figure 5. a), b): Reevaluation of Lung Tomography. Please, improve the quality of the figure.
Response 3 and 4: We have changed them according to the recommendations, now the images from the figures are provided directly by the Radiology Service where the patient was admitted for investigations, it is the best quality we have available.
Point 5: Discussion and Conclusions: L346-351. Please, summarise here the novelty of the study.
Response 5: We tried to achieve this requirement in point 6, in the conclusions.
Point 6: L440-444. Please, summarise here the novelty of this paper and the possible clinical implications.
Response 6: We tried to reach points 5 and 6 in the conclusions, L482-488, accordingly to your valuable advise.
“Thus, to emphasize the impact of interstitial lung involvement in systemic scleroderma, we have presented a narrative review and a case from a patient admitted to our Pneumology Department from Mures Clinical County Hospital which were diagnosed with pulmonary fibrosis related to systemic scleroderma. By understanding the mechanism of developing progressive pulmonary fibrosis related to systemic scleroderma, a personalized treatment can pe established to increase the patient’s outcome and by this to increase the quality of life.”
As you suggested minor editing of English language used in this article, the authors have tried their best to correct the errors as per required.
We greatly appreciate the time you have taken to engage with our work. We hope that the revised manuscript meets the above requirements and we look forward to your feedback.
With great respect,
The Authors

Reviewer 2 Report
Comments and Suggestions for Authors
Dear Authors, I have read your manuscript with interest.
The current manuscript titled: "Multidisciplinary Aproach: A Goal For The Management Of Complications In Systemic Scleroderma. Literature Review And Case Scenario" represents an important analysis of evolving field of Rheumatology and Immunology.
In my opinion, these are the adjustments which should be made to increase the value of your manuscript:
1. In Introduction chapter, please, add the review aim and and specify that this is a narrative review. Also, add information about SSc’ epidemiology.
2. In clinical case, please describe in more detail patient’ clinical general examination and laboratory analysis.
3. Lines 207-212: it is recommended to remove words with capital letters from the text and make them the same as the general text.
4. It is recommended to remove tables 1 and 3 from the manuscript, since their description is contained in the text and they do not provide any important additional information.
5. It is recommended to divide the Discussions and Conclusions sections into 2 separate sections. Also, in Discussions chapter add information about the importance of vitamin D in SSc and consider this holistic review https://doi.org/10.1155/2021/9782994.
6. The manuscript contains some punctuation errors, please revise the text.
Comments on the Quality of English LanguageMinor editing of English language required
Author Response
Dear Reviewers, thank you very much for taking the valuable time to review our work and for providing your feedback, which I find very useful in significantly improving our article. The authors have carefully noted all coments and guidance and we tried our best to provide point-by-point answers to your observations along with the obligatory explanations. We have highlighted the changes within the manuscript via Track-Changes.
Response to Reviewer 2 Comments:
Point 1: In Introduction chapter, please, add the review aim and and specify that this is a narrative review. Also, add information about SSc’ epidemiology.
Response 1: Thank you for your useful indications, and as a result, we added larger informations about our paper in L42-44 and details regarding epidemiology of systemic sclerosis from L46 to 50.
“In this paper, we have done a case-based literature review of the pulmonary fibrosis related to systemic scleroderma, because the increased risk of developing pulmonary fibrosis and pulmonary artery hypertension is among the leading causes of SSc-related death.
Systemic sclerosis is a rare rheumatological condition, with a prevalence of 7.2–33.9, 13.5–44.3 and 8.2 per 100,000 individuals in Europe, North America and United Kingdom, respectively with an annual incidence of 0.6–5.6 per 100,000 individuals. Like many other rheumatological conditions, women are more commonly affected than men, with a reported ratio of between 3:1 to 8:1 [2,6].”
Point 2: In clinical case, please describe in more detail patient’ clinical general examination and laboratory analysis.
Response 2: We tried our best to highlight all of the modifications discovered in our patients' clinical examinations, but on account of your valuable suggestions, we have improved this subchapter by including a table with the modified values of laboratory analysis.
|
Parameter |
Values |
|
White blood cells |
4630/µl |
|
Hemoglobin |
12,5 g/dl |
|
Hematocrit |
38,8% |
|
Platelets |
254000/µl |
|
Uric acid |
303 µmol/L |
|
Alanine transaminase |
12 U/L |
|
Aspartate aminotransferase |
20 U/L |
|
Total cholesterol |
4,37 mmol/l |
|
Creatinine |
0,82 mg/dl |
|
Alkaline phosphatase |
210 U/L |
|
Glucose |
124 mg/dl |
|
Lactate dehydrogenase |
233 U/L |
|
C-reactive protein |
5,13 MG/L |
|
Fibrinogen |
345 mg/dl |
|
Total proteins |
7,2 g/dl |
|
Anti-Scl-70 antibodies |
7.2 |
|
Antinuclear antibodies |
25.3 UI/ml |
Point 3: Lines 207-212: it is recommended to remove words with capital letters from the text and make them the same as the general text.
Response 3: We made them the same as the general text.
Point 4: It is recommended to remove tables 1 and 3 from the manuscript, since their description is contained in the text and they do not provide any important additional information.
Response 4: We appreciate your point of view very much, but we would like, if it is possible, that tables 1 and 3 to remain in this paper, as they contain all the necessary parameters for the respiratory tests described, and only the modified values and the conclusions are described in the text.
Point 5: It is recommended to divide the Discussions and Conclusions sections into 2 separate sections. Also, in Discussions chapter add information about the importance of vitamin D in SSc and consider this holistic review https://doi.org/10.1155/2021/9782994.
Response 5: According to your recommendation, we separated the two sections. Furthermore, we are appreciative for the suggested reference, a really fascinating paper work from which we gathered significant material for our article, as can be seen in lines 429-434.
“Moreover, vitamin D may interfere with each of the pathobiological processes triggered in SSc, including autoimmunity, peripheral vasculopathy, and fibrosis, due to its immunomodulatory, cardioprotective, and antifibrotic biological actions. Vitamin D levels appear to be considerably reduced as compared to healthy controls, and vitamin D supplementation is mandatory [75].”
Point 6: The manuscript contains some punctuation errors, please revise the text.
Response 6: We revised and edited the manuscript and hope that all punctuation errors were fixed.As you suggested minor editing of English language used in this article, the authors have tried their best to correct the errors as per required.
We greatly appreciate the time you have taken to engage with our work. We hope that the revised manuscript meets the above requirements and we look forward to your feedback.
With great respect,
The Authors

Round 2
Reviewer 1 Report
Comments and Suggestions for Authors
The manuscript has been improved. No further comments
Comments on the Quality of English LanguageMinor changes of English language are required
Reviewer 2 Report
Comments and Suggestions for Authors
I agree with the changes made, which significantly improve the quality of the manuscript.
Comments on the Quality of English LanguageMinor editing of English language required